# Seasonal Variations in the Physical Fitness of South African Premier Soccer League Players over an Annual Training Macrocycle (Nine Months)

**DOI:** 10.3390/jfmk10010038

**Published:** 2025-01-21

**Authors:** Mduduzi Rhini, Robert Charles Hickner, Rowena Naidoo, Takshita Sookan-Kassie

**Affiliations:** 1Discipline of Biokinetics, Exercise and Leisure Sciences, School of Health Sciences, University of KwaZulu-Natal, University Road Westville Private Bag X54001, Durban 4000, South Africa; rhickner@fsu.edu (R.C.H.); naidoor3@ukzn.ac.za (R.N.); sookan@ukzn.ac.za (T.S.-K.); 2Department of Health, Nutrition and Food Sciences, College of Education, Health and Human Sciences, Florida State University, Tallahassee, FL 32306, USA

**Keywords:** fitness, professional players, seasonal variation, soccer

## Abstract

**Background**: Anecdotal data indicate that the physical fitness of soccer players fluctuates across the season. This is often a concern for coaches, since players are expected to be at optimal fitness during matches on weekly basis across the season. **Objectives**: To analyze the physical fitness variation in South African Premier Soccer League players over an annual training macrocycle. **Methods**: Twenty-four Premier Soccer League players belonging to the same team participated in the study. Players went through fitness assessments at three stages of the season: at the beginning of pre-season (T1); mid-first round in-season (T2); and mid-second round in-season (T3). The assessments included body fat percentage; sit and reach; vertical jump; 10 and 30 m sprints; and YoYo Intermittent Recovery Level 2 (YoYo IR2). **Results**: There was a significant increase in body fat percentage from T1 to T2 (*p* < 0.001), and a slight decline was evident at T3 (*p* = 0.04). Flexibility was significantly greater at T2 (*p* < 0.001) compared to T1 and T3. Vertical jump significantly improved at T3 (*p* = 0.004) compared to T1 and T2. A similar trend was evident in the YoYo IR2, where players reached the highest levels at T3 (*p* < 0.001). However, there were no significant changes in the 10 and 30 m sprints across the season. **Conclusions**: These results indicate that, indeed, some parameters, such as body fat percentage and flexibility, are likely to fluctuate as the season progresses. However, it is also evident that a gradual improvement can be achieved, as seen in vertical jump and YoYo IR2.

## 1. Introduction

In soccer, maintaining optimal physical fitness is essential for players to perform high-quality actions consistently throughout weekly matches. However, achieving and sustaining peak physical fitness across a season presents a significant challenge [1,2]. Research has shown that player fitness levels often fluctuate over the course of a season rather than following a steady, progressive improvement [3]. For example, while it is generally expected that components like body fat percentage will decrease towards the end of pre-season or as the season progresses—due to the typically high training load of pre-season—this is not always the case. The available fitness data on soccer players typically indicate a steady decline in body fat percentage from pre-season to the in-season [4,5], while some show no significant changes in body composition throughout the season [6]. These variations could be influenced by changes in training load, diet, match fitness, or the lack thereof [7].

Seasonal variations in other fitness parameters, such as power and speed, have also been explored, and indications of improvements from pre-season to in-season are reported [3,6]. Meckel et al. [8] found that vertical jump performance significantly improved by the end of pre-season from 37.0 cm to 39.0 cm in elite players, though no further improvements were noted during the season. The pre-season strength and plyometric training was credited for the initial gains, while the absence of strength training during the in-season likely accounted for the lack of further progress. Importantly, vertical jump performance was maintained, suggesting that the training load was sufficient to preserve pre-season gains even though continued improvement would have been ideal. This is critical, as power directly affects speed and overall soccer performance.

In another study, Dragijsky et al. [2] investigated agility and linear speed in the Czech league on four occasions during the season. Their findings indicated significant improvements across all tests, with 30 m linear speed being notably higher during the competitive period (4.98 s) compared to both pre-season (5.15 s) and at the start of the season (5.07 s). These findings align with previous research showing that 10 and 30 m sprint performance improved significantly as the competitive season progressed [9]. These mid-season improvements may be attributed to an appropriate training load with continual high-intensity exercises.

In terms of aerobic fitness, normative data suggest that it tends to increase steadily from pre-season to mid-season before plateauing or declining towards the end of the season [8]. For instance, a study on professional Greek National Championship players found that maximal oxygen uptake (VO_2max_) significantly declined towards the end of the season from 57.96 mL/kg/min to 56.07 mL/kg/min [1]. Similarly, Fessi et al. [6] reported a decline of approximately 3% in aerobic fitness during in-season, which was attributed to the reduced training load during in-season. According to the authors, the weekly training load during in-season was not appropriate to maintain performance. This appears to be the challenge with most strength and conditioning coaches across the literature, whereby, in an attempt to avoid accumulated fatigue, the training load often is reduced to the point where it is not sufficient enough to maintain the gains made through pre-season.

Although there is available literature on fitness variations in soccer players, it is worth noting that there is currently limited data coming from the African region. Therefore, the aim of this study was to investigate the physical fitness variations in South African Premier Soccer League players over an annual training macrocycle, ranging from pre-season until the second round in-season. Based on the reviewed literature, it is hypothesized that body composition measurements will decrease as the season progresses, whereas, speed, power, and aerobic capacity would increase and then plateau towards the end of the season.

## 2. Materials and Methods

### 2.1. Population and Sample

This was an observational study where twenty-four players belonging to the same professional team participated in the study. The team competed in the Premier Soccer League (PSL) in South Africa, which is the highest division. Data were collected during the 2019–2020 season.

### 2.2. Ethical Considerations

All players agreed to participate and gave their written, informed consent in accordance with the Helsinki Declaration. The University of KwaZulu-Natal’s Biomedical Research Ethics Committee also approved the study protocol and granted ethical clearance BE695/18.

### 2.3. Procedures

Fitness testing was administered on three (3) occasions across the season, as indicated in Table 1. The first assessment (T1) was carried out in the second week of pre-season; the second one (T2) was during the first round in-season in October (20 weeks apart), and the last one (T3) was during the second round in-season in March (21 weeks apart). By including the pre-season period in this study, the team’s season was nine months long. A total of 26 official games were played during this study, with 23 being league matches and three being tournament (cup/knockout competitions) matches. The team had already played seven games by the time the T2 assessments were carried out and 26 games when the T3 assessments were conducted. Only players who were injury free and had participated on all three occasions were included. Players who missed one or two testing sessions due to illness, injury, or participating in international duties were excluded from the study.

Generally, the team had five training sessions in a normal week, in addition to one match, as shown in Table 2.

Tuesday (or the first day of the training week)—Double sessions, with strength sessions taking place in the morning and field conditioning in the afternoon. A typical strength session would include at least (1) three lower body exercises, e.g., back squat, the Romanian deadlift, and lunges; (2) one core exercise, e.g., plank; (3) one upper body exercise, e.g., shoulder press. The afternoon conditioning session would mainly be small-sided games (4v4; 5v5; or 6v6) with at least four sets of 7–12 min.Wednesday—Technical/tactical session; coaches used this session to introduce the game preparation for the upcoming match. Would normally include warm-up, passing drills, and medium small-sided games (7v7; 8v8) with task constraints and specific tactical demands.Thursday—Tactical session, including a warm-up, possession game, 11v11, and offensive set plays.Friday—An activation session, which would normally be an hour-long session, with mobility, reaction speed/fun game, rondo, attacking patterns/schemes, and defensive set plays.Saturday—Match DaySunday—Recovery session for the players who played more than 60 min and maintenance session for the players who played less and those who did not make the squad. The recovery session would include mobility and fun games, whereas the maintenance session included conditioning small-sided games.Monday—Complete rest day.

Testing was conducted on two (2) separate days, starting at 09:00 in the morning on both days. Non-fatiguing tests to examine body fat percentage, flexibility, and vertical jump were administered on the first day. The second day was dedicated to the 10 and 30 m sprints and the YoYo IR. The tests were preceded by a warm-up which was conducted by the strength and conditioning (S and C) coach of the reference team. Well-trained interns and postgraduate students from the discipline of Biokinetics, Exercise, and Leisure Sciences at the University of KwaZulu-Natal assisted with taking the measurements and conducting the tests. The assessments were carried out by the same research team at the same venue, starting at the same time, and the order of tests was the same across the season. Although the players were professional players and were familiar with fitness testing, the researchers explained the test instructions in detail prior to every test. This was performed to ensure the reliability and quality of the test. The American College of Sports Medicine (ACSM) guidelines for exercise testing and prescription were used for the measurements.

### 2.4. Body Fat Percentage (%)

Players were instructed to wear only their training shorts, and skinfold measurements were taken with the player standing upright at seven (7) sites: subscapular; triceps; chest; midaxillary; suprailiac; abdomen; and thigh. Plastic (SlimGlides) calipers were used, and the generalized seven-site skinfold formula was used to estimate body fat percentages [10]. All measurements were taken on the right side of the body. Duplicate measurements were taken at each site, and an average of the two measurements was recorded, provided they were within 1 to 2 mm. A retest was conducted if duplicate measurements were not within 1 to 2 mm.

### 2.5. Flexibility—Sit and Reach

A sit and reach test (cm) was performed to assess the hamstring, trunk, and hip joint flexibility. A standard baseline sit and reach box was used to perform the test. The sit and reach test is considered to be a reliable alternative for the assessment of hamstring extensibility with a moderate (r = 0.70 to 0.76) mean correlation coefficient [11,12]. Players removed their shoes and sat on the floor with their legs extended and feet placed directly against the box. The players were asked to place their hands on top of each other and slowly reach forward as far as possible to the yardstick on top of the box, then hold for three seconds. The distance reached was then recorded. The players were given three (3) tries, and the best score was recorded for analysis.

### 2.6. Vertical Jump

A standing vertical jump test (cm) was performed to evaluate the players’ lower-limb explosive power in a vertical direction, following the protocol of De Salles et al. [13]. The vertical jump test has been validated as a reliable method of measurement of the explosive power of the lower limbs of athletes, with a correlation coefficient of between 0.76 and 0.80 [13,14]. Players had to wear their light training kits with trainers, and all players performed a standard warm-up before the test. To perform the test, players were instructed to stand, with feet approximately shoulder-width apart. A zero starting position was established by having players reach the highest point on the wall while in an erect standing position. From there, players moved into a semi-squat position, with flexed knees, hips, and ankles, and jumped to the highest point on the wall. Each player had three tries, and the highest score was recorded. All tests were performed at an indoor basketball court with a wooden floor.

### 2.7. The 10 M and 30 M Sprints

Ten- and 30 M sprints (seconds) were used to assess the players’ speed, with the 10 M sprint assessing mostly the acceleration, based on the protocol used by Chamari et al. [15]. The players completed a dynamic warm-up with the S and C coach before the test. A two-point start was applied in both sprints, and measurements were taken using the Brower Timing Systems (Draper, UT, USA) electronic timing gates. The Brower Timing System has been used widely in high-performance sports and in research to assess linear speed, and its validity and reliability (coefficient of variation = 3.1, 1.8, 2.0, and 1.3%) have been reported [16,17]. Players had two attempts in each sprint, and the better performance was recorded for analysis. All the tests were performed on an outdoor soccer field.

### 2.8. YoYo Intermittent Recovery Test Level 2 (YoYo-IR2)

Aerobic capacity was assessed using the YoYo IR2 (m), which is one of the most soccer-specific tests available and has been considered reliable and valid [18,19]. The test consists of 2 × 20 m runs back and forth, with five meters behind the finish line used for a recovery walk between each 2 × 20 m run. Players were required to run 20 m and then turn and run another 20 m back while keeping up with a series of beeps. The test was designed to progressively increase speed at each level, and players had a 10 s recovery period after every shuttle. The highest level reached and the total distance run (m) were recorded. Players were given a warning if they had (i) missed the beep, (ii) did not touch the line, or (iii) did not wait for the beep. Each player was given only two (2) warnings, and, on the third warning, the player was considered out and the highest level they had completed was recorded for analysis. The test was terminated once all the players were considered out. The test was performed on an outdoor soccer field, and players used their normal soccer boots during the test.

### 2.9. Data Analysis

All data analyses were conducted using the statistical software SPSS, version 25 (IBM, Chicago, IL, USA). Descriptive statistics were presented as the means ± standard error of means (SEM). A repeated measurements ANOVA was applied to test for the differences in fitness characteristics between the three fitness testing times. Partial eta-squared (ηp2) values were calculated to determine effect size. The level of significance was set at *p* < 0.05. If the result was significant, a Tukey’s post hoc analysis was carried out to determine specific differences between the stages in the season, and also between playing positions.

## 3. Results

Table 3 shows the anthropometric and physical characteristics of the players across the season. A significant difference was evident in body fat percentage across the season. Analysis showed that body fat percentage at T2 (7.0 ± 0.3%, *p* < 0.001) was significantly higher compared to both T1 (5.2 ± 0.2, *p* < 0.001) and T3 (6.4 ± 0.3%, *p* = 0.04). In addition, the body fat percentage values were significantly higher in T3 (*p* = 0.01) than in T1. A significant decline (F = 12.41, *p* < 0.001, ηp^2^ = 0.35) was evident in flexibility (sit and reach) in T3 (34.3 ± 1.3 cm, *p* < 0.001), compared to T3 (40.6 ± 1.4 cm). In terms of speed, there were no significant differences in either the 10 M or 30 M sprints (*p* > 0.05) across the season. However, it can be noted that the lowest speeds for both the 10 M and 30 M sprints were observed during T1. Vertical jump values were lower during the pre-season period (T1 = 49.2 ± 1.1 cm), with a significant improvement (F = 6.36, *p* = 0.004, ηp^2^ = 0.26) during the competition period, in T3 (52.9 ± 1.4 cm). Similarly, the YoYo IR2 (F = 17.59, *p* < 0.001, ηp^2^ = 0.54) results were best during T3 (870.0 ± 45.6 m) compared to T1 (582.5 ± 29.2 m) and T2 (750.0 ± 44.1 m). Figure 1 shows the physical characteristics of players according to their playing positions.

## 4. Discussion

This study aimed to analyze the seasonal variations in physical fitness among professional soccer players across three stages of an annual training macrocycle. The main findings indicated that, while some fitness components significantly improved throughout the season, others either fluctuated or showed no consistent progression. Notably, body fat percentage and flexibility fluctuated over the season, whereas vertical jump performance and aerobic capacity demonstrated steady, progressive improvements. These variations could be attributed to factors such as training methods, changes in training load, and game time (or the lack thereof).

The findings regarding body fat percentage were particularly intriguing. The lowest body fat percentage was recorded at the start of pre-season, as players returned from the off-season, and the highest values were observed during the first round of in-season, with a slight decrease being evident in the second round. These findings contrast with most of the literature on professional soccer players [4,20,21]. Generally, players are expected to accumulate body fat during the off-season due to the reduced training load and lack of competition [5,22]. This is attested by previous research conducted by Caldwell and Peters [23] and Magal et al. [9] where they revealed that, in their studies, body fat percentage was highest before pre-season compared to the other four stages of the season, with the lowest values being reported at the end of the season. The increase in body fat percentage between T1 and T2 in the present study, despite the rigorous pre-season training, was particularly unexpected. The pre-season period is regarded as the most intensive training time in soccer, and it is characterized by high levels of aerobic, anaerobic, and strength training activities, which are directly associated with body fat percentage reductions [21]. A plausible explanation for this unexpected finding might be the changes in living arrangements for certain players during the off-season leading to altered dietary habits. Thus, it is recommended that future studies should attempt to monitor the caloric intake and expenditure of the players across the season to gain a clearer understanding of the changes in body composition.

Concerning flexibility, there was a significant improvement between T1 and T2, followed by a significant decline from T2 to T3. The initial improvement could be attributed to the pre-season conditioning program, which emphasized athletic development and muscular fitness, aligning with findings by Caldwell and Peters [23]. They suggested that flexibility improvements during the first round of the season are due to pre-season training programs that emphasize mobility. Meckel et al. [8] similarly reported increased flexibility during pre-season, with reductions during the off-season due to a lack of stretching and movement.

With respect to power, a significant increase in vertical jump performance was observed between T1 and T2, with the highest improvement occurring in T3. These findings contrast with the previous literature, which suggests that vertical jump performance typically improves during pre-season but stagnates or declines during in-season [6,8]. The improvement in vertical jump in the present study could be attributed to the team’s regular strength and power training, which was conducted once a week throughout the season. Surprisingly though, these power gains did not translate into significant improvements in linear speed. There were no substantial changes in the 10 and 30 m sprints despite the well-established relationship between explosive power and speed [24]. This inconsistency aligns with the findings by Chamari et al. [15] who found no significant correlation between vertical jump height and sprint performance in young elite soccer players. The distinction between jump height and peak velocity is crucial, as the latter may have a stronger relationship with sprint performance. This is in agreement with findings reported by Kulakowski et al. [25] where they found no significant relationship between jump height and any sprint test.

Aerobic capacity showed a significant improvement over the season, which is consistent with previous research [8,9]. The players’ performance in the YoYo IR2 test improved by over 20% between T1 and T2, with peak performance being observed at T3. This suggests that the coaching staff successfully implemented an appropriate training load during the in-season, primarily through small-sided games. The findings align with Dragijsky et al. [2], who observed significant endurance improvements toward the end of the season. However, other studies have shown that players often reach their maximum aerobic capacity during pre-season and either maintain or experience a decline during the season [20,23]. The significant decline in the other studies could be attributed to the fact that the competition period is usually longer and that, once players reach their maximum levels during pre-season, thereafter it becomes difficult to further improve during the season [2].

Thus, monitoring the seasonal changes in physical fitness and performance throughout the season is imperative to ensuring that players participate at their optimal level. Coaches have to be able to control and maintain an appropriate training load to keep players’ physical fitness at an optimal level throughout the season. With the season lasting for almost 11 months, coupled with injuries and congested fixtures, actively monitoring player readiness and recovery becomes paramount. In some cases, the squad is divided into different groups, e.g., starters, non-starters, and injured/return from injury, which often creates a challenge. However, proper periodization, with clear objectives and outcomes where training is individualized and tailored according to the players’ needs, can alleviate the problem. An ideal situation is one where the players’ fitness progressively increases throughout the season and does not plateau or decline, as this would have an effect on performance and, ultimately, on the match outcome.

## 5. Conclusions

In conclusion, this study’s results show that players can improve certain parameters during the long competitive soccer season. With the exceptions of body composition, flexibility, and speed, the improvements in power and aerobic fitness indicate that the in-season training program employed by the coaches provided an appropriate physical stimulus for superior adaptation. The variation in body fat percentage could be the result of multiple factors, such as changes in personal life or home arrangements or changes in training load (from pre-season to in-season). The lack of improvement in linear speed is surprising; however, the lack of linear speed training in soccer, in general, may have contributed.

Therefore, it is recommended that future studies should consider situational and contextual factors, such as playing time, training load, diet, and injuries, when investigating seasonal fitness in soccer players. Additionally, it would also be beneficial if testing is conducted at the beginning and at the end of each stage of the season, e.g., at the beginning of pre-season and at the end of pre-season.

## 6. Limitations

This study was subject to some limitations. Firstly, the COVID-19 pandemic prevented final testing at the end of the season, which was a significant limitation. Testing at the end of the season would have provided a more comprehensive perspective on seasonal fitness variations. Secondly, testing was conducted at the beginning of pre-season and during in-season, but no tests were carried out at the end of pre-season or the start of in-season, limiting insights into the complete fitness progression. Finally, this study did not adequately account for contextual factors such as injuries, playing time (starters vs. non-starters), and match performance data. These limitations restrict the study’s ability to draw broad conclusions. A more comprehensive analysis of these factors in future research could enhance the reliability and relevance of the results.

## 7. Practical Applications

The continual improvement in power and aerobic fitness throughout the season in the present study indicates that, with appropriate periodization and consistent monitoring, fluctuation and decline in fitness can be avoided. Awareness and understanding of the various factors that influence fitness are important for coaches to be able to prescribe appropriate and tailor-made training loads for individual players.

## Figures and Tables

**Figure 1 jfmk-10-00038-f001:**
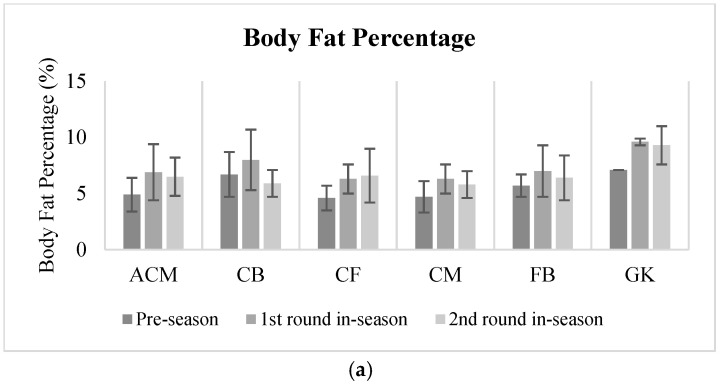
Physical characteristics of players per playing position across the three stages of the season (pre-season; 1st round in-season and 2nd round in-season). Fitness tests: (**a**) Body fat percentage (%); (**b**) Sit and reach (cm); (**c**) Vertical jump (cm); (**d**) YoYo Intermittent Recovery Test Level 2 (m). Playing positions: ACM—attacking central midfielders; CB—center backs; CF—center forwards; CM—central midfielders; FB—full backs; GK—goalkeepers.

**Table 1 jfmk-10-00038-t001:** Physical fitness testing timeline.

Physical Fitness Testing Timeline			
	Pre-Season	First Round In-Season	Second Round In-season
Month	July	October	March
Testing Time	T1—start of pre-season	T2—first round in-season	T3—second round in-season
Matches played	None	Seven (7) matches	25 matches

Note: T1—first testing time (second week of pre-season); T2—second testing time 0 weeks later T3—third testing time (21 weeks later).

**Table 2 jfmk-10-00038-t002:** Weekly training plan/schedule.

In-Season Weekly Training Plan/Schedule							
	Monday	Tuesday	Wednesday	Thursday	Friday	Saturday	Sunday
Morning 9.00 a.m.	OFF	Gym: Strength Session	Tactical Session		Activation Session and Defensive Set Plays		Recovery Session and Maintenance Session
Afternoon 2.00 p.m.		Field Conditioning Session		Tactical Session and Offensive Set Plays		Match Day	

**Table 3 jfmk-10-00038-t003:** Mean *±* SEM anthropometric and physical fitness characteristics of players across the three stages of the season (pre-season; first round in-season; and second round in-season).

Players’ Physical Characteristics Across the Season						
		N	Pre-Season (T1)	1st Round In-Season (T2)	2nd Round In-Season (T3)	*p*-Value: Post Hoc
Body Composition	BMI (kg/m^2^)	24	23.1 ± 0.3	23.0 ± 0.3	22.9 ± 0.3	*p* > 0.5
	BF (%)	24	5.2 ± 0.2 *	7.0 ± 0.3 *	6.4 ± 0.3 ^§^	0.001: T1 < T2; T1 < T3
Flexibility	Sit and Reach (cm)	24	36.9 ± 1.7	40.6 ± 1.4	34.3 ± 1.3 ^#^	0.001: T1 < T2; T2 > T3
Power	Vertical Jump (cm)	19	49.2 ± 1.1 *	50.3 ± 1.3 *	52.9 ± 1.4 ^§^	0.004: T1 < T2; T2 < T3
Speed	10 m Sprint (s)	18	1.7 ± 0.1	1.7 ± 0.1	1.8 ± 0.0	*p* > 0.5
	30 m Sprint (s)	18	4.1 ± 0.1	4.1 ± 0.1	4.2 ± 0.0	*p* > 0.5
Aerobic Capacity	YoYo-IR2 (m)	16	582.5 ± 29.2	750.0 ± 44.1 *	870.0 ± 45.6 ^#^	0.001: T1 < T2; T2 < T3
	Speed Level		14.5 ± 0.7	18.7 ± 1.1	21.7 ± 1.1	

Note: Results are expressed as means ± SEM. BMI—body mass index. BF%—body fat percentage. YoYo IR2—YoYo Intermittent Recovery Test Level 2. The significance level was set at *p* < 0.05. * denotes significance between T1 and. T2; ^§^ between T1 and T3; and ^#^ between T2 and T3.

## Data Availability

All data sets used and analyzed in this study will be made available by the corresponding author upon reasonable request.

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
