# Peer review of "Seasonal Variations in the Physical Fitness of South African Premier Soccer League Players over an Annual Training Macrocycle (Nine Months)"

_jfmk, 2025, doi:10.3390/jfmk10010038_

Round 1
Reviewer 1 Report
Comments and Suggestions for Authors
I would like to thank the editor for the opportunity to review this manuscript. The study provides a well-written and insightful analysis of seasonal variations in physical fitness parameters among high-level soccer players. The authors present data from three distinct time points during a competitive season, offering a valuable perspective on performance development. However, I have identified several major and minor concerns that require attention before the manuscript can be considered for publication.
Major:
Inter-Individual Variability and Contextual Factors
- The analysis treats data from 24 players as independent observations, without consideration for contextual factors that could influence individual performance. While the manuscript outlines the team’s typical training schedules, it does not account for inter-individual variations in training load across the season.
- Injuries: Injury exposure, a common issue in professional soccer, was not addressed in the analysis. Mid- and long-term injuries can induce detraining effects, potentially skewing group results. The authors should provide injury data, either as a grouping variable or a covariate, to improve the statistical model’s robustness.
- Playing Time: Variations in match exposure (e.g., starting vs. non-starting players) can significantly impact seasonal performance trends. Incorporating playing time or playing status as a confounding variable in sub-analyses would enhance the contextual interpretation of the findings (e.g., see Nobari et al.).
Validity and Reliability of Performance Tests
- The manuscript lacks detailed references on the validity and reliability of the performance tests used, particularly for the jump-and-reach test, where learning effects might influence results. Reliable methods, such as the MyJump App, could be considered in future research (e.g., https://www.nature.com/articles/s41598-023-46935-x).
- Adding references to the reliability of all field tests performed would provide readers with a clearer understanding of the diagnostic quality.
Temporal Limitations in Testing
- While the limitations section acknowledges the timing of testing, the absence of a fourth time point between T2 and T3 reduces the study's validity. Although this cannot be rectified for the current manuscript, the authors should emphasize this as a significant limitation.
Minor:
Introduction
- While the general issue and performance subscales are well-introduced, clear hypotheses for each outcome are missing and should be articulated based on existing evidence.
Emphasize the relevance and novelty of the study to strengthen its positioning.
Methods
Provide additional contextual information, such as injury and match exposure data, to improve group characterization.
Include references on the validity and reliability of all performance tests conducted.
Specify whether the athletes were familiar with the testing procedures or performing them for the first time.
Report on outdoor testing conditions, particularly climate variability, as it could influence performance outcomes (e.g., https://pmc.ncbi.nlm.nih.gov/articles/PMC8677617/).
Results
Clarify abbreviations for playing positions in Figure 1 and explain why positional comparisons were not included in earlier analyses.
Provide a plot showing individual performance changes over the season to offer a more granular perspective.
Recheck the standard deviations in the third measurement point of Figure 1, as they appear erroneous.
Discussion
Add a concluding section that integrates findings within the broader context of performance monitoring. Highlight how the observations could assist in optimizing training strategies. Specifically, address how contextual variables, such as injury status and playing time, could provide more actionable insights.
Reviewer 2 Report
Comments and Suggestions for Authors
The authors provide an investigation into potential differences in fitness over time in elite soccer players. While this is an interesting study, several clarifications are required which would strengthen the overall submission.
1) Abstract – remove the description of the abstract (line 14).
2) Comment throughout – “data” are plural, revise tenses when applicable accordingly.
3) Line 65 – the abbreviation “VO2max” has not yet been introduced in your text – please include the term for which VO2max stands.
4) The data included in the “population and sample” section (Lines 76-77) belong in the Results section of the manuscript – the Materials and Methods should be strictly a recounting of what was performed.
5) Were the assessments at all time points for any given athlete performed by the same trained technicians? If not, what is the potential for inter-rater reliability to introduce error into these data?
6) Methodologically, for body fat percentage – what was done if the duplicate measures of skinfold at any given site were more than 2mm apart?
7) Table 2 is included in the manuscript, but no reference to is made throughout the text of the paper. Additional information is provided to provide context about this table – is this weekly training plan/schedule consistent throughout the entire season? Do intensities, durations, etc. change throughout the season? How might these factors impact the fitness measurements taken at each study time point?
8) Line 168 – “Significant” should have a capital S.
9) Lines 168-173 – the several results provided here indicate “p=0.00.” It is customary to present the actual p-values associated with these results. This can be “p<0.01” if appropriate – but p-values will never be 0.00.
10) In Table 3, what are the units for “Speed Level”? Further, what is the explanation for the reduced sample size for the Power, Speed, and Aerobic Capacity tests? Additionally, please report standard deviation as these data are representative of a single soccer team over one season, which is not representative of the population.
11) For Figure 1, how many players were in each position group? What do these error bars represent (standard error or standard deviation)? Since these data are presented by time point and by position group – were any two-way ANOVAs performed to determine if there were differential responses across time by position group? If there was logic to support this presentation of data in this format, one would assume that this was analyzed for potential interactions.
12) Lines 252-254 – the nutritional status/dietary habits of these players are a critical component. How should future studies standardize this to limit any confounding effects of these changes, to better shed light on impacts of training and soccer-specific effects on body fat percentage?
Reviewer 3 Report
Comments and Suggestions for Authors
Title: Indicate the number of monitored weeks, as in football the microcycle has little relevance given the weekly competition schedule (or even within a shorter interval/microcycle).
Abstract: The presented results should be supported by descriptive and inferential statistics.
Introduction:
- Include the quantitative values of the physical fitness variables analyzed, considering previous published studies.
- The specific mention of the Greek National Championship is unclear, given that the South African Premier Soccer League is being analyzed.
- The objectives should specify which dimensions of physical fitness are being evaluated. Additionally, clarify what is meant by "annual training macrocycle." Does this refer to a sports season?
Materials and Methods:
- The Population and Sample section only describes the participants; however, since physical fitness evaluations are being conducted, it is essential to describe the number of observations/evaluations performed for each player.
- Please include the inclusion and exclusion criteria. Also, describe the sample size power calculation.
- Specify the materials used to collect each physical fitness variable. Detail how each test was conducted, which variables were collected, and their respective units of measurement.
- In Table 2, the Weekly Training Plan/Schedule is too general. Specify what is done in each session. More importantly, explain how and when the physical fitness evaluations were conducted.
Results:
- Table 3 presents only the Mean ± SEM; it would be important to also include the inferential statistics from the calculated ANOVA.
- Include values in the figures where significant differences exist. Additionally, comparisons by field positions should be contextualized in the methodology and objectives of the study.
Discussion:
- Expand the discussion and deepen the comparison of the results with previously published studies.
- Add practical applications, study limitations, and future perspectives.
Conclusion: Focus on the key outcome in an objective manner.
References: The references do not follow the instructions for authors.
Comments on the Quality of English LanguagePlease, improve.
Round 2
Reviewer 2 Report
Comments and Suggestions for Authors
The authors greatly increased the strength of the article and are to be commended for this work.
Reviewer 3 Report
Comments and Suggestions for Authors
Dear authors,
In the title, remove ‘(nine months)’; replace it with ‘during an official season’. A suggestion for the title of the manuscript would be: "Seasonal variations in the physical fitness of South African Premier Soccer League players during an official season". In football, the macrocycle has little meaning.
In addition, the p-value alone means nothing. I recommend that the authors report the F statistics, the reported effect size (eta squared) and the mean differences and significances reported for the post hocs.
The table 2 presents the Weekly Training Plan/Schedule more clearly.However, the text on lines 132 to 152 should be set out in narrative form and backed up with literature on which they based the terminology of the training exercises (e.g. small-sided games, rondo, body exercises, etc.).
Well done!
Comments on the Quality of English LanguagePlease, improve.